# Service Design of a Loss Prevention Device for Older Adults with Dementia

**DOI:** 10.3390/geriatrics8050093

**Published:** 2023-09-15

**Authors:** Cheng-Kun Hsu, Cheng-Chang Liu, Tung Chang, Jing-Jing Liao, Chi-Min Shu

**Affiliations:** 1Ph.D. Program in Design, Chung Yuan Christian University, Taoyuan 320314, Taiwan; ookunoo@gmail.com; 2Department of Visual Communication Design, Ling Tung University, Taichung 408284, Taiwan; g9632722@yuntech.org.tw; 3Doctoral Program, Graduate School of Engineering Science and Technology, National Yunlin University of Science and Technology (YunTech), Yunlin 64002, Taiwan; d10010023@yuntech.org.tw; 4Department of Business Administration, TransWorld University, Yunlin 64002, Taiwan; j876985@gmail.com; 5Department of Safety Health, and Environmental Engineering, National Yunlin University of Science and Technology (YunTech), Yunlin 64002, Taiwan

**Keywords:** smart alert bracelet, elderly dementia, wearable device, degree of dementia, convenience sampling

## Abstract

This aim of this research was to explore the appraisal of the use of smart alert bracelets by older adults diagnosed with dementia. Convenience sampling was adopted to recruit older adults with dementia in Yunlin County, Taiwan. A manual questionnaire survey was conducted, and SPSS 26.0 statistical software was used for analysis. The results of this study showed noticeable positive correlation results in the post-test for the modes “wearing device”, “degree of dementia”, and “field configuration”. Based on the experimental results, the following suggestions are provided: (1) in terms of statistical calculation, the statistical results were affected by changes in some participants; (2) as for the design of equipment, to be more suitable for adult use, the size and color of bracelets need to be optimized; (3) as for the problem of battery charging of the device, because the charging location of the device is not easy to find, it is better to extend device standby time; (4) regarding the selection of equipment, older adults with early-stage dementia could be concerned about the function of the wearable device, so it is recommended to provide a device designed with clear functions, such as a watch, so that older adults are more willing to wear it. Patients diagnosed with moderate and severe dementia should be advised to use concealed non-sensory devices, such as charms and cards, to better facilitate assistance from caregivers in wearing them; and (5) as for the device, in case of a loss event, in addition to mobile phone notifications, other light and sound device notifications can be added, allowing caregivers to pay more attention to information in real time. In summary, the feedback from caregivers and older adults suggests that if the device is to be used without charging, the overall design should be light and small, which is more suitable for service designs.

## 1. Introduction

According to World Health Organization (WHO) statistics, in 2021, there were more than 55 million people with dementia in the world, and this number is expected to grow to 139 million people by 2050. Dementia-related costs are USD 1.3 trillion per year, and are likely to reach USD 2.8 trillion by 2030 due to increased care costs [1].

According to the results derived from a epidemiological survey on dementia conducted by the Taiwan Alzheimer’s Disease Association (TADA) [2] commissioned by the Ministry of Health and Welfare in 2011, and demographic data released by the Ministry of the Interior in September, 2016, Taiwan’s dementia population has exceeded 260,000, and among them, 243,430 dementia patients were aged over 65 as of 2016. Simply stated, approximately 1 in every 100 people suffer from dementia. On this basis, in another 25 years, it is estimated that there will be nearly 670,000 people diagnosed with this chronic disease. This implies that 3 out of every 100 people could possibly suffer from dementia in the future [3]. Practically speaking, dementia has become a prevailing health problem of increasing concern in all countries. Taiwan is not immune from this, and the growing prevalence of dementia will become an emerging issue in family and social care in the future.

### 1.1. Motivation

Developed countries such as the USA, Japan, and other countries in Europe have already paid special attention to social issues related to aging from the perspective of normalization and social welfare [4]. Right now, the Taiwanese government is vigorously promoting local care-related policies, such as the Long-term Care Plan 2.0. Specifically speaking, if caregivers can fully understand relevant information and adequately use the plan’s services and technologies, it can substantially abate the physical and mental burden of caregivers.

In actual care, if technology-related devices can be used to assist, in addition to lessening the pressure on caregivers in one way or another, they can also gradually alter caregivers’ interpersonal interactions and the way they care for each other [5]. Despite this, when technological devices are employed as a carrier to capture relevant physiological or behavioral information platforms (such as cloud platforms), it is rather necessary to pay attention to personal privacy issues [6]. Relevant devices can be roughly divided into invasive devices, e.g., implanted chips and QR code tattoos, or those designed as part of wearable devices, such as bracelets, necklaces, and clothing. At present, the aim of this study is to use wearable devices as an auxiliary tool to better avoid the use of controversial and intrusive items. The purpose of doing so is to not only diminish the care pressure caused by patients’ cognitive impairment [7], but also to provide comprehensive discussions about service design platforms for older adults diagnosed with dementia, caregivers, wearable device technology assistance, and care. To that end, the characteristics of the physical and mental functions of older adults diagnosed with dementia, the interactions and lifestyles of older adults diagnosed with dementia and their caregivers and the service designed (with care referred to as the core goal and smart life as the integration goal) were considered and constructed through the technology care network platform [8], which is the main focus of this study.

### 1.2. Purpose

Because of the physical and mental deterioration of older adults diagnosed with dementia and the actual care needs of caregivers, some major design issues are associated with technology devices that can meet the relevant technological needs of users; thus, it is necessary not only to make devices easier for patients and caregivers to operate, but also to compensate, through technology, for older adults suffering from deterioration of their physical and mental functions.

In this study, attempts are made to provide discussions about the interpersonal interactions and lifestyles of older adults diagnosed with dementia, with consideration given to the interactive care network system constructed by smart life. In doing so, it is hoped that a service design with care as the core will be provided, thereby release the physiological burden of institutional caregivers and meeting psychological needs. According to the World Alzheimer Report [9], current clinics are inadequate to cope with huge demands. If improvement is made in the division of labor in grassroots and community medical care, we may increase the diagnosis rate of dementia and therefore reduce medical expenditure. In addition, we may better provide for the diversified needs of older adults diagnosed with dementia. In Taiwan, as in many countries in Asia, the main place where patients diagnosed with dementia are given care is the home of the patient or that of his/her relatives; more than 70% of the caregivers are unitary [10], and the caregivers are mostly their spouses [11]. Because of the fact that the nursing care system in Taiwan is not perfectly and fully developed, the development of the mechanism in this regard is relatively slow. Due to the low birthrate in Taiwan, there has been an inevitable trend of caregivers becoming older; older adult caregivers are required to take care of older adults diagnosed with dementia.

“Caregiving for the patient” is not an easy task, for there are differences between modes of care in the East and the West [12]. In Western society, when they become old, patients can choose to live in nursing institutions for the rest of their life. Meanwhile, in Eastern societies, older adults may leave home to live in care institutions, and these patients may be accused of unfilial piety [13].

Home care is an effective way to provide symptomatic relief for people with dementia in a familiar environment [14]. However, from the perspective of the physician, the family, or the caregiver, caregiving for an elderly dementia patient is a relatively difficult task. In the process of caregiving, the peripheral symptoms of older adults’ dementia—such as wandering (getting lost or going missing)—often become the greatest source of psychological pressure for caregivers in the process of caregiving for or finding patients [15].

If there is a complete set of methods or service platforms [16], non-professionals, such as family members or on-site caregivers, can better improve their awareness of dementia prevention and care [17,18], and can therefore understand how to assess, use information, and adjust their own model of care, in addition to improving and enhancing the quality of care for dementia patients. Said methods or platforms can also effectively and efficiently release the psychological pressure faced by most caregivers.

## 2. Research Framework

This study was mainly based on a service design designed for older adults diagnosed with dementia. This was to explore how to use the technology and interpret the information it produces, so as to achieve the goal of caring for older adults suffering from dementia and their caregivers. In doing so, they can achieve the purpose of caregiving.

“Older adults diagnosed with dementia” is the main research topic. The focus of this research is on the three main aspects of “caregivers”, “wearable devices”, and “care platforms” to better explore the correlation between them. In this study, we used a first-person approach to the problem. In so doing, we created and designed the concept and proposed the product to older adults diagnosed with dementia for testing and development. Moreover, we connected the existing relevant system platforms and technologies and integrated a care platform with service design at the core, as shown in Figure 1.

### 2.1. Literature Review

As guided by “service design”, this study provides the discussions of the psychological and physical needs of the older adults with dementia and caregivers from the perspectives of “local ageing” and “shared value”, so as to better achieve the benefits of caregiving. A review of the related literature mainly includes the following three main aspects: 2.1.1, Caregivers and platforms for caregiving; 2.1.2, Wearable device technology and platforms for caregiving; and 2.1.3, Caregivers and wearable device technology.

#### 2.1.1. Caregivers and Platform for Caregiving

(1)What is dementia?

According to the Diagnostic and Statistical Manual of Mental Disorders, DSM-5/2013, dementia (a major neurocognitive disorder), mainly includes attention, executive function, learning and memory, language (conversation), the activities of daily living, social cognition for understanding the moods or thoughts of others, and other functions; at least one of these declines, and the independence of daily life also declines [19].

(1)Also, the focus is placed on the behavioral characteristics of people diagnosed with dementia; i.e., central symptoms versus peripheral symptoms.

Dementia refers to cognitive dysfunction with memory impairment, and these central symptoms, such as forgetfulness, loss of orientation, and judgment impairment, are due to cognitive dysfunction. Its symptoms can vary, and this solely depends on the psychological state caused by the patient’s personality, temperament, surrounding environment, and interpersonal relationships, including hallucinations, wandering, violence, incontinence, personality changes, and other symptoms. There are several stages of dementia, and various typical symptoms and behaviors, as provided by the Alzheimer’s Association [20] and listed in Table 1 below.

#### 2.1.2. Wearable Device Technology and Platforms for Caregiving

Individuals living with dementia experience progressive and irreversible loss of memory and other cognitive functions, together with motor and balance impairments [21]. Older adults diagnosed with dementia, if not properly cared for, may become another concern for social security services [22]. Regarding home safety and accidental disappearance, most elderly dementia patients are unable to sleep well at night. For caregivers, the problem of disappearance caused by wandering often recurs, and they do not know when it will occur, or when the person diagnosed with dementia is out in the middle of the night; therefore, caregivers are often deprived of sleep. This causes them to suffer from physical and mental exhaustion, and this becomes a source of great physical and mental stress for caregivers [23].

To better solve the problems faced by these patients and alleviate pressure on caregivers, caregivers can use wearable devices that have sensing functions to record and evaluate the information obtained, and therefore to establish relevant mechanisms and systems to maintain dementia patients’ safety at home and outside.

#### 2.1.3. Caregivers and Wearable Device Technology

Wearable technology is integrated with existing technology to develop miniature and lightweight devices so that they can be worn on people’s bodies to capture relevant information, such as physiological information; then, data can be transmitted to the cloud platform through wired or wireless communication. In the meantime, the results can be delivered to personal mobile devices, so that their users can acquire the information they need [24,25].

## 3. Study Background

### 3.1. Current Situation and Trend of the Aging Elderly Population in Taiwan and Yunlin

According to the internal affairs statistics circular of the Department of Household Registration (MOI) in 2020, about 3.6 million older adults over 65 years old are registered, accounting for 15.28% of the total population. This means that Taiwan is now classed as an “aged society”, as listed in Table 2.

The proportion of older adults is increasing year by year. As registered in Taiwan at the end of 2019, the proportion of older adults in the population was relatively high in Chiayi County, making up 19.68%. In Yunlin County, they accounted for 18.52%, and in Taipei City, they accounted for 18.07%. According to the National Development Council, it is expected that the population of the older adults in Taiwan will exceed 20% in 2026, meaning Tiawan will be classified as a super-aged society along with Japan, South Korea, Singapore, and some European countries.

The problems and concerns arising from aged societies and the increase in the support rate for the older adults will emerge one after another. They are described as follows.

First of all, regarding the impact of the older adults: older adults generally suffer from discrimination because they are prone to diseases, and high demand for medical resources leads to increasing expenditure, inadequate institutional care, mental decline, and psychological aging. Furthermore, if some older adults suffer from dementia, they lose the ability to live independently and become part of the dependent population. As far as the impact of family is concerned, the ratio of older adult caregivers has increased. Young adults have become old, and they have to take care of elderly adults alongside raising the young, and so the pressure on them has increased sharply. If there is a dementia or bedridden elderly person in the family, the physical and psychological health of the family caregiver can suffer, and the family’s living standards, work, and rest may become disordered. Finally, there is also an impact on the community impact; the increase in the older adult population in the community requires more manpower and material resources to protect and care for older adults, for example, through setting up day care institutions and nursing care institutions for older adults, providing more services to assist in finding the lost elderly (e.g., patrol teams), and caring for older adults living alone. 

### 3.2. Aged Society and the Needs of Dementia Patients

From the above-mentioned information, it can be seen that the proportion of the elderly population in Yunlin County, Taiwan, is higher than that of the national average. This suggests that the number of older adults diagnosed with dementia is most of all the the counties and cities.

According to the online questionnaire survey conducted by the Ministry of Health and Welfare on the seven strategies for dementia prevention and care policy 2.0 in November 2017, among the 2228 responses, the top three policies that the public most wanted to focus on were “support and assistance for the caregivers of families with dementia”, “public awareness and friendly attitudes towards dementia”, and “dementia diagnosis, treatment, and care”. “Support and assistance for caregivers (whether they are family members or non-family members) of families with dementia” is of high importance and so considered the first priority.

Therefore, the purpose of study is fourfold: (1) to achieve the purpose of supporting family caregivers and helping to achieve the goal of zero loss of dementia/non-dementia elderly people in Yunlin County, Taiwan; (2) to use the application of smart alert bracelets to enable families and institutions to know exactly where the older adult is at home at any time by accurately knowing the location of the older adult with dementia/ordinary elderly person; (3) to set up a warning area through a safety electronic fence in order that the older adults diagnosed with dementia or the ordinary elderly can move freely and walk around near their homes; and (4) to maintain a higher quality of life and safety and enable family members not to worry about the problem of whether older adults will become lost.

## 4. Applicable Conditions and Applicable Scope

The attempt made in this study is to provide devices designed with intelligent anti-loss Bluetooth positioning technology and position sensors that can be used to identify the location of the older adults at any time and reduce the occurrence of incidents of becoming lost.

### 4.1. Applicable Conditions

The intelligent anti-loss device provided by our team has the following advantages:Low power consumption; it can be used continuously for more than 12 months without charging.The device is lightweight and convenient, weighs less than 25 g, and is easy to wear.The sensing range can be extended to indoor and outdoor areas, and the distance can be set according to the user’s needs.The cost of the device is not high, and its overall features are better than those of any other products on the market.

### 4.2. Application Scope

The status, functions, and application scope of GPS technology include:Precise detection, which is widely used in observatories, communication system base stations, and TV stations.Engineering construction: GPS equipment is widely used for surveying in the construction of roads, bridges, and tunnels.Exploration and mapping: GPS equipment is used in both field exploration and urban planning.Navigation:(1)Weapon navigation: Precision-guided munition, and cruise missiles;(2)Vehicle navigation: Vehicle dispatching and monitoring systems;(3)Ship navigation: Ocean navigation, and port/inland water diversion;(4)Aircraft navigation: Route navigation, approach, and landing control;(5)Interstellar navigation: Satellite positioning;(6)Personal navigation: Personal travel and wild exploration.Positioning:(1)Vehicle anti-theft devices;(2)Mobile phones, PDA, PPC, other mobile anti-theft equipment, electronic maps, and positioning systems;(3)Anti-loss systems for children and special groups;(4)Precision agriculture: Agricultural machinery navigation, automatic driving, and high-precision land leveling;(5)Timing: Used to provide accurate synchronous clocks for telecommunication base stations, TV transmission stations, etc.

### 4.3. Six Features of GPS Technology

Not affected by any weather conditions;Global coverage, up to 98%;High-precision three-dimensional fixed points, fixed speed, and timing;Fast, time-saving, and efficient;Wide range of applications, and multiple functions;Mobile positioning.

## 5. Research Discussion

The aim of this study was to discuss the decision-making factors involved in design of the dementia smart alert bracelet. The purpose of doing so is as follows:To provide discussion about the definitions and types of dementia smart alert bracelets;To provide discussion about the selection decisions of dementia smart alert bracelets;To provide discussion about the factors influencing the form of dementia smart alert bracelets.

It is hoped that the results derived from this study can provide practical references for the Yunlin County Government, Taiwan, and other counties and cities with aging problems to observe. Also, to better achieve the above research objectives, this study intended to adopt the questionnaire survey method, based on relevant literature discussions; through expert correction after the compilation of questionnaires, the development of questionnaires was established.

## 6. Experiments and Hypotheses

### 6.1. Experiments

This study aimed to explore the different decision-making factors involved in the design of a dementia smart protection bracelet. To achieve the above research purpose, the experimental framework of this study is as follows (Figure 2):

### 6.2. Hypotheses

To achieve the purpose of this research, based on the content of the literature review and the research framework, the proposed research hypotheses are provided as follows:

**H1.** *There are significant differences between the older adults usability experience in terms of “wearing device”, “degree of dementia”, and “field setting”*. 

**H2.** *The “degree of dementia” has a positive relationship with “usability experience”*.

**H3.** *The “wearing device” has a positive relationship with “usability experience”*.

**H4.** *The “field setting” has a positive relationship with “usability experience”*.

## 7. Subjects

The proportion of the older adults is increasing year by year. As registered in Taiwan at the end of 2019, the proportion of the population made up of older adults was relatively high in Chiayi County, making up 19.68%. In Yunlin County, they accounted for 18.52%, and in Taipei City, they accounted for 18.07%. According to the National Development Council, it is expected that the population of the older adults in Taiwan will have exceeded 20% in the 2026, meaning it is classified as a super-aged society along with Japan, South Korea, Singapore, and some European countries.

This project is based on an experimental research design. This study heavily relies on interview protocols and evaluation reports as the main work items. A day care center located in Yunlin County, Taiwan, was selected as the experimental site. The experiments were carried out with five anti-loss devices to test the effectiveness and willingness of wearers to wear the related devices. The test subjects were provided by the institution, and the number of test participants was expected to be 100. The suitable anti-lost device was created ad designed according to the evaluation of the researcher and the willingness of the test subjects.

## 8. Materials

This questionnaire was compiled after a comprehensive literature review of relevant topics published in a number of research reports and discussions with a supervisor. After the pre-test questionnaire, SPSS software was used to analyze and delete the items. In this study, a self-listed questionnaire was used as the survey tool and measured using a Likert scale (5-point option); the questionnaire was divided into three parts: personal background information, experimental field information, and usability experience information. The instructions were provided as follows.

### 8.1. Personal Background Information

The divisions of the personal background information of this questionnaire can be roughly described as follows:Gender was divided into male and female, with a total of two options.The participants were divided into the following different age ranges: 55–60 years old, 61–65 years old, 66–70 years old, 71–75 years old, 76–80 years old, 81–85 years old, 86–90 years old, 91–95 years old, and 96 years old and above, making a total of nine options.The participants were divided in to the following different education levels: primary education, lower secondary education, upper secondary education (vocational school), Bachelor’s education, tertiary education, literate, and illiterate, making a total of seven options.The participants were asked whether they live with their children or not, and there were two options: “Yes” or “No”.Degree of dementia was rated on a scale: CDR: 0.5, CDR: 1, CDR: 2, and CDR: 3, with a total of four options.The participants were asked whether they have children or not, and there were two options: “Yes” or “No”.

### 8.2. Experimental Information

The experimental information of this questionnaire is roughly divided as follows.

In the experimental field, the districts are divided into the following seven options:Douliu “Little Sun” Day Care Center for older adults;Gukeng “Little Sun” Day Care Center for older adults;“Bulaotang” Dongshi Day Care Service Center;“Bulaotang” Taixi Day Care Service Center;“Chang Tai Old School” Douli Day Care Center;“Chang Tai Old School” Huwei Day Care Center;“Chang Tai Old School” Baozhong Day Care Center.

Wearable Device

A total of five items were used. They are listed as follows: a Bluetooth card (card), a Bluetooth bracelet (orange wristband), a Bluetooth charm (charm), a GPS bracelet (pink wristband), and a GPS watch (black watch), as provided in Table 3.

### 8.3. Usability Experience

The usability questions are divided into three categories: Safety, comfort, and convenience, with a total of ten questions.

Simple ↔ Difficult.Relaxed ↔ Oppressive.Convenient ↔ Inconvenient.Stable ↔ Tense.Reliable ↔ Unreliable.Safe ↔ Dangerous.Useful ↔ Difficult to use.Lightweight ↔ Bulky.Effortless ↔ Laborious.Comfortable ↔ Uncomfortable.

## 9. Implementation Research

The aim of this study was to discuss the evaluation of the usability of smart protective bracelets for older adults with dementia. A questionnaire survey was conducted for older adults with dementia in Yunlin County, Taiwan. The implementation design of this study were carried out in the following stages:

### 9.1. The First Stage

Based on the research objectives and a review of the relevant literature, the integration of the concept of usability with age-friendly design, the research, and the questionnaire design were carried out. Informed consent was obtained from the day care centers subject to Yunlin County Government, Taiwan, and all of the participants were willing to wear and install Bluetooth receivers.

### 9.2. The Second Stage

According to the compiled questionnaire, the survey (pre-test) was conducted, and the sampling part was first set to the subjects who had not used the relevant devices. The distribution experiment of the first device was performed and introduced, and the distribution time was from 4 February 2020 to 5 March 2020. Furthermore, with the assistance of The Yunlin County Government, Taiwan, the device was distributed to the day care centers with their consent. An explanation of the background login and operation methods of the device was given by a social worker at the day care center. From 16 March 2020 to 14 April 2020, 100 questionnaires were distributed, and 84 valid questionnaires were returned.

### 9.3. The Third Stage

This section that follows analyzes the returned questionnaires and provides corrections of the related questionnaire questions. Afterwards, the revised and modified questionnaires were used to carry out the second stage of the survey research. For the target subject and the location of the distribution, the older adults diagnosed with dementia were recruited from day care centers overseen by the Yunlin County Government, Taiwan. During the second phase, equipment testing was performed from 16 March 2020 to 14 April 2020. From 4 May 2020 to 8 June 2020, 100 questionnaires were sent out, and 75 valid questionnaires were returned.

### 9.4. The Fourth Stage

After the questionnaires were gleaned, the data were subject to a cross-comparison analysis of the variables in the designed questionnaire; SPSS (version 26.0 statistical software) was used to conduct the analysis. This was to determine whether the disguised phases would affect each other, so as to draw further conclusions and make suggestions based on the results of the analysis.

## 10. Research Methods and Data Analysis

To better achieve the purpose of the research, according to the research framework, the research hypothesis was mainly based on information collected from the participants, who were recruited from Yunlin County, Taiwan. This was to explore the impact of the evaluation and gain a better understanding of the usability of the alert bracelet and its subsequent effect on the decision made by the older adults diagnosed with dementia to wear it. In addition, this was also to formulate relevant improvement strategies, provide industry references, and achieve the purpose of marketing promotion.

## 11. Research Results and Analysis


*Convenience Sampling*


This study adopted the method of convenience sampling to distribute questionnaires, and after the questionnaires were collected, SPSS software was used to conduct a cross-comparison analysis of the options in the questionnaires.

To better understand the purpose of questionnaire and the reliability and validity of the questionnaire, the following items were used as the basis of its construction:The correlation coefficient between each question and the total score reached a significance level of 0.01.The correlation coefficient between each question and its total score reached a significance level of 0.001.According to the total score of the subscale, the samples were divided into high and low groups (the highest and the lowest were 27% respectively), and the mean difference between the high and low groups was compared item by item, and the *t*-value reached a significance level of 0.01 [26].

### 11.1. Statistical Analysis of the Personal Background Data of the Subjects

To glean the personal background and information of the respondents, and to understand the differences in the test intervals and regions of the smart alert bracelet, this section deals with a background analysis of the interviewees, and conducts a statistical analysis of the number and percentage of gender, age, education level, test site, whether there are children, living together, dementia, and devices worn.

This study was divided into two experiments:A.The relevant information and content of the first-time subjects are detailed below, and the main reasons for the differences are described as follows.(1)Most of the subjects were women (67.9%).(2)The age distribution was mainly 81–85 years old (30.29%), followed by 86–90 years old (17.86%), and 76–80 years old (16.67%).(3)The general education background of the subjects is elementary school (42.86%), followed by junior high school (28.57%).(4)All the subjects had children, but 15.5% of the subjects did not live with their children.(5)In terms of the degree of dementia, most of the elders had moderate dementia, 45.2% of the subjects were moderately so, 28.6% were mild, and 16.7% were severe.(6)The reasons for the reduction in the number of tested personnel are as follows: (a) emotional issues (one person); (b) unable to answer properly (one person); (c) family members decided to quit (two people); (d) long-term leave (six people); and (e) those who left the day care center (six people).B.The following points related to the information and content of the respondents to the second questionnaire and the main reasons for the changes were mainly considered to describe the differences and distributions. However, upon the second questionnaire, due to the personal issues of the respondents, the number of responses decreased. The details of said responses are as follows.C.In the second experiment, there were fewer male and female subjects. The majority of subjects were still women (66.67%).(1)Their age was mainly between 81–85 years old (38.7%), followed by 86–90 years old (20.0%), and then 76–80 years old (14.7%).(2)The most common subjects had an elementary school education (37.3%), followed by a middle school education (30.7%).(3)All the subjects had children, but 15.5% of the subjects did not live with their children.(4)In terms of the degree of dementia, most of the elders had moderate dementia, 45.2% of the subjects were moderately so, 28.6% were mild, and 16.7% were severe.(5)Except for the older adults who had withdrawn from the first stage, the reasons for the withdrawal of nine people from the second test are given as follows: (a) the relatively short wearing time (three people); (b) the long-term intervals or interruptions (three people); and (c) their admission from the day care center (three people), along with other unidentified factors.

### 11.2. Analysis of Different Background Variables in Each Piece of Research

This section mainly provides discussions about the differences in research into “wearing devices”, “degree of dementia”, and “usability experience”, among the variables of different personal backgrounds of the older adults.

Therefore, in this study, an independent sample *t*-test was employed to test whether there were significant differences in the research variables of “gender”, “whether they live together”, “whether they have children”, and “wearing devices”. If there was a noticeable difference between “field settings”, “wearing devices”, “degree of dementia”, and “usability experience”, the Scheffe method or the Tukey HSD method was further used to conduct post-mortem testing.

### 11.3. Analysis Independent Samples t-Test

In this study, an independent sample t-test was used to test the differences in the research variables among older adults with different “gender”, “whether they live together”, “whether they have children”, and “wearing devices”. The study found that there was no significant difference in each research variable. Although there was no significant difference, it can be seen from the results obtained from the two stages of experiment that male subjects were more likely to wear protective bracelets.

The main differences revealed in the results are stated as follows (Table 4 and Table 5):(1)“Simple”: For wearing protective bracelets based on gender, the feedback collected from men after their wearing the device for the first time was that it was difficult to wear the device. After they were used to wearing the device, the preference for this increased (favorability increased to 2.64%).(2)“Lightweight”: In terms of lightness, the feeling of the device ranged from repulsion and lack of familiarity to greater familiarity; the device’s weight was actually similar to the weight of jewelry worn in the past, so women’s favorability increased (2.14%, increasing to 2.48%). Men generally did not have the habit of wearing accessories, so there was no increase in favorability (2.19%, increasing to 2.28%).(3)“Effort-saving”: In terms of energy-saving, the feelings of men did not change (2.37% increased to 2.44%). It was also found through interviews that after getting used to wearing the device, the sense of weight was reduced, so women felt that the device’s weight was quite acceptable (2.35% increased to 2.44%).

### 11.4. One Way ANOVA

This study used an analysis of variance (ANOVA) to examine the differences in research variables among these older adults diagnosed with different “degrees of dementia”, “wearing devices”, and “field settings”, using the above-mentioned research methods for the analysis of data. If the results of the analysis showed a significant difference, the Scheffe method or Tukey’s HSD method was used for post hoc comparisons. The study found that there was no significant difference in all of the research variables except the variable of “field setting”.

#### 11.4.1. Analysis of Different “Wearable Devices” and Usability Experience

In this study, two tests were conducted to compare differences. The test results of the pre-test (the first test) were compared with “Bluetooth bracelet (orange wristband)”, “GPS bracelet (pink wristband)”, and “GPS watch (black watch)”, where these three models had better feedback. These results were found to be contradictory to those of the actual interview. Furthermore, the reason for this part of the result changed according to the test field and the degree of dementia. In the interviews, the Bluetooth card (cards) and Bluetooth charms (charms) had optimal evaluations. This difference was based on the present, and the overall weight of the cards and charms was relatively light, so there was indeed a difference.

The test results of the post-test (the second test) included “GPS Watch (black watch)”, “Bluetooth bracelet (orange wristband)”, and “Bluetooth Charm (charm)” with high feedback. The reason for this result may be that the older adults had relevant wearing experience and were less repulsed. After the wearing habits of the pre-test, the testees developed curiosity, comprehension ability, and sense of habit in wearing the device, which caused the differences in the research results.

##### Wearing Results of the First Test (Table 6)

(1)Bluetooth wristbands (orange wristbands) had a higher sense of “Relaxed”, “Reliable”, and “Safe” for the older adults.(2)GPS wristbands (pink wristbands) had a higher sense of “Convenient”, ”Simple” and “Stable” for the older adults.(3)GPS watches (black watches) had a higher sense of “Lightweight”, “Useful”, and “Effortless”.(4)Bluetooth wristbands (orange wristbands) and GPS wristbands (pink wristband) showed equal status in the two indicators of “Simple” and “Relaxed”.

**Table 6 geriatrics-08-00093-t006:** The first experiment’s descriptive statistics for usability experience.

		*N*	Average Value	Standard Deviation	Standard Error of Mean	Confidence Interval, CI		Minimum	Maximum
						Lower Limit	Upper Limit		
Usability experience “Simple”	Bluetooth card	15	2.07	0.594	0.153	1.74	2.40	1	3
	Bluetooth bracelet	15	2.47	0.640	0.165	2.11	2.82	2	4
	Bluetooth pendant	19	2.26	0.653	0.150	1.95	2.58	1	4
	GPS wrist	19	2.47	1.020	0.234	1.98	2.97	1	5
	GPS watch	16	2.25	0.447	0.112	2.01	2.49	2	3
	total	84	2.31	0.711	0.078	2.16	2.46	1	5
Usability experience “Relaxed”	Bluetooth card	15	2.13	0.640	0.165	1.78	2.49	1	4
	Bluetooth bracelet	15	2.53	1.060	0.274	1.95	3.12	1	5
	Bluetooth pendant	19	2.05	0.405	0.093	1.86	2.25	1	3
	GPS wrist	19	2.53	1.020	0.234	2.03	3.02	1	5
	GPS watch	16	2.38	0.719	0.180	1.99	2.76	1	4
	total	84	2.32	0.809	0.088	2.15	2.50	1	5
Usability experience “Convenient”	Bluetooth card	15	2.40	0.632	0.163	2.05	2.75	1	3
	Bluetooth bracelet	15	2.47	0.743	0.192	2.06	2.88	1	4
	Bluetooth pendant	19	2.32	0.749	0.172	1.95	2.68	1	4
	GPS wrist	19	2.63	0.831	0.191	2.23	3.03	1	4
	GPS watch	16	2.56	0.814	0.203	2.13	3.00	1	4
	total	84	2.48	0.752	0.082	2.31	2.64	1	4
Usability experience “Stable”	Bluetooth card	15	2.33	0.617	0.159	1.99	2.68	1	3
	Bluetooth bracelet	15	2.27	0.704	0.182	1.88	2.66	1	3
	Bluetooth pendant	19	2.26	0.562	0.129	1.99	2.53	1	3
	GPS wrist	19	2.37	0.955	0.219	1.91	2.83	1	4
	GPS watch	16	2.25	0.683	0.171	1.89	2.61	1	3
	total	84	2.30	0.708	0.077	2.14	2.45	1	4
Usability experience “Reliable”	Bluetooth card	15	2.47	0.743	0.192	2.06	2.88	1	4
	Bluetooth bracelet	15	2.73	0.458	0.118	2.48	2.99	2	3
	Bluetooth pendant	19	2.68	0.478	0.110	2.45	2.91	2	3
	GPS wrist	19	2.53	0.697	0.160	2.19	2.86	1	3
	GPS watch	16	2.56	0.512	0.128	2.29	2.84	2	3
	total	84	2.60	0.583	0.064	2.47	2.72	1	4
Usability experience “Safe”	Bluetooth card	15	2.53	0.743	0.192	2.12	2.94	1	4
	Bluetooth bracelet	15	2.67	0.488	0.126	2.40	2.94	2	3
	Bluetooth pendant	19	2.47	0.612	0.140	2.18	2.77	1	3
	GPS wrist	19	2.42	0.769	0.176	2.05	2.79	1	4
	GPS watch	16	2.38	0.719	0.180	1.99	2.76	1	3
	total	84	2.49	0.668	0.073	2.34	2.63	1	4
Usability experience “Useful”	Bluetooth card	15	2.33	0.724	0.187	1.93	2.73	1	4
	Bluetooth bracelet	15	2.67	0.617	0.159	2.32	3.01	2	4
	Bluetooth pendant	19	2.47	0.612	0.140	2.18	2.77	1	3
	GPS wrist	19	2.37	0.761	0.175	2.00	2.74	1	4
	GPS watch	16	2.69	0.602	0.151	2.37	3.01	2	4
	total	84	2.50	0.668	0.073	2.36	2.64	1	4
Usability experience “Lightweight”	Bluetooth card	15	2.07	0.704	0.182	1.68	2.46	1	4
	Bluetooth bracelet	15	2.00	0.535	0.138	1.70	2.30	1	3
	Bluetooth pendant	19	2.00	0.667	0.153	1.68	2.32	1	3
	GPS wrist	19	2.21	0.918	0.211	1.77	2.65	1	4
	GPS watch	16	2.50	1.155	0.289	1.88	3.12	1	5
	total	84	2.15	0.829	0.090	1.97	2.33	1	5
Usability experience “Effortless”	Bluetooth card	15	2.13	0.640	0.165	1.78	2.49	1	3
	Bluetooth bracelet	15	2.47	0.834	0.215	2.00	2.93	1	4
	Bluetooth pendant	19	2.16	0.765	0.175	1.79	2.53	1	4
	GPS wrist	19	2.26	0.653	0.150	1.95	2.58	1	3
	GPS watch	16	2.81	0.911	0.228	2.33	3.30	1	4
	total	84	2.36	0.786	0.086	2.19	2.53	1	4
Usability experience “Comfortable”	Bluetooth card	15	2.33	0.724	0.187	1.93	2.73	1	4
	Bluetooth bracelet	15	2.73	1.100	0.284	2.12	3.34	1	5
	Bluetooth pendant	19	2.11	0.567	0.130	1.83	2.38	1	3
	GPS wrist	19	2.37	0.955	0.219	1.91	2.83	1	4
	GPS watch	16	2.63	0.957	0.239	2.11	3.14	1	5
	total	84	2.42	0.881	0.096	2.23	2.61	1	5

##### Wearing Results of the Second Test (Table 7)

(1)Bluetooth wristbands (orange wristbands) had a higher sense of “Convenient” and “Stable”, “Reliable”, “Useful”, and “Effortless”.(2)GPS watches (black watches) had a higher sense of “Simple”, “Relaxed”, “Lightweight”, and “Comfortable”.(3)Bluetooth pendants (straps) had a higher sense of “Safe” for the older adults.(4)After multiple comparisons of the wearable devices, it was found that the “Lightweight” use experience of the GPS watches (black watches) prompted more significant and positive feedback than Bluetooth cards (cards) (Table 8).

**Table 7 geriatrics-08-00093-t007:** The second experiment’s descriptive statistics for usability experience.

		*N*	Average Value	Standard Deviation	Standard Error of Mean	Confidence Interval, CI		Minimum	Maximum
						Lower Limit	Upper Limit		
Usability experience “Simple”	Bluetooth card	19	2.42	0.692	0.159	2.09	2.75	1	4
	Bluetooth bracelet	12	2.67	0.651	0.188	2.25	3.08	2	4
	Bluetooth pendant	14	2.36	0.929	0.248	1.82	2.89	1	5
	GPS wrist	15	2.40	0.632	0.163	2.05	2.75	2	4
	GPS watch	15	2.73	0.704	0.182	2.34	3.12	2	4
	Total	75	2.51	0.724	0.084	2.34	2.67	1	5
Usability experience “Relaxed”	Bluetooth card	19	2.58	0.838	0.192	2.18	2.98	1	4
	Bluetooth bracelet	12	2.58	0.669	0.193	2.16	3.01	2	4
	Bluetooth pendant	14	2.43	0.756	0.202	1.99	2.87	2	4
	GPS wrist	15	2.27	0.704	0.182	1.88	2.66	1	4
	GPS watch	15	2.73	0.799	0.206	2.29	3.18	2	4
	Total	75	2.52	0.760	0.088	2.35	2.69	1	4
Usability experience “Convenient”	Bluetooth card	19	2.53	0.841	0.193	2.12	2.93	1	4
	Bluetooth bracelet	12	2.75	0.452	0.131	2.46	3.04	2	3
	Bluetooth pendant	14	2.64	0.633	0.169	2.28	3.01	2	4
	GPS wrist	15	2.60	0.737	0.190	2.19	3.01	1	4
	GPS watch	15	2.73	0.704	0.182	2.34	3.12	2	4
	Total	75	2.64	0.690	0.080	2.48	2.80	1	4
Usability experience “Stable”	Bluetooth card	19	2.58	0.507	0.116	2.33	2.82	2	3
	Bluetooth bracelet	12	2.92	0.289	0.083	2.73	3.10	2	3
	Bluetooth pendant	14	2.86	0.363	0.097	2.65	3.07	2	3
	GPS wrist	15	2.73	0.458	0.118	2.48	2.99	2	3
	GPS watch	15	2.80	0.414	0.107	2.57	3.03	2	3
	Total	75	2.76	0.430	0.050	2.66	2.86	2	3
Usability experience “Reliable”	Bluetooth card	19	2.63	0.496	0.114	2.39	2.87	2	3
	Bluetooth bracelet	12	2.92	0.289	0.083	2.73	3.10	2	3
	Bluetooth pendant	14	2.86	0.363	0.097	2.65	3.07	2	3
	GPS wrist	15	2.73	0.458	0.118	2.48	2.99	2	3
	GPS watch	15	2.87	0.352	0.091	2.67	3.06	2	3
	Total	75	2.79	0.412	0.048	2.69	2.88	2	3
Usability experience “Safe”	Bluetooth card	19	2.63	0.496	0.114	2.39	2.87	2	3
	Bluetooth bracelet	12	2.83	0.389	0.112	2.59	3.08	2	3
	Bluetooth pendant	14	2.86	0.363	0.097	2.65	3.07	2	3
	GPS wrist	15	2.73	0.458	0.118	2.48	2.99	2	3
	GPS watch	15	2.80	0.414	0.107	2.57	3.03	2	3
	Total	75	2.76	0.430	0.050	2.66	2.86	2	3
Usability experience “Useful”	Bluetooth card	19	2.74	0.653	0.150	2.42	3.05	2	4
	Bluetooth bracelet	12	3.00	0.603	0.174	2.62	3.38	2	4
	Bluetooth pendant	14	2.71	0.469	0.125	2.44	2.98	2	3
	GPS wrist	15	2.60	0.507	0.131	2.32	2.88	2	3
	GPS watch	15	2.87	0.516	0.133	2.58	3.15	2	4
	Total	75	2.77	0.559	0.065	2.64	2.90	2	4
Usability experience “Lightweight”	Bluetooth card	19	2.16	0.602	0.138	1.87	2.45	1	3
	Bluetooth bracelet	12	2.42	0.515	0.149	2.09	2.74	2	3
	Bluetooth pendant	14	2.29	0.611	0.163	1.93	2.64	1	3
	GPS wrist	15	2.33	0.617	0.159	1.99	2.68	2	4
	GPS watch	15	2.93	0.961	0.248	2.40	3.47	2	5
	Total	75	2.41	0.718	0.083	2.25	2.58	1	5
Usability experience “Effortless”	Bluetooth card	19	2.26	0.562	0.129	1.99	2.53	1	3
	Bluetooth bracelet	12	2.58	0.515	0.149	2.26	2.91	2	3
	Bluetooth pendant	14	2.29	0.611	0.163	1.93	2.64	1	3
	GPS wrist	15	2.40	0.632	0.163	2.05	2.75	2	4
	GPS watch	15	2.73	0.884	0.228	2.24	3.22	1	4
	Total	75	2.44	0.663	0.077	2.29	2.59	1	4
Usability experience “Comfortable”	Bluetooth card	19	2.47	0.841	0.193	2.07	2.88	1	4
	Bluetooth bracelet	12	2.58	0.669	0.193	2.16	3.01	2	4
	Bluetooth pendant	14	2.43	1.016	0.272	1.84	3.02	1	5
	GPS wrist	15	2.20	0.561	0.145	1.89	2.51	2	4
	GPS watch	15	2.87	0.990	0.256	2.32	3.42	1	4
	Total	75	2.51	0.844	0.097	2.31	2.70	1	5

**Table 8 geriatrics-08-00093-t008:** The second experiment’s post hoc multiple comparison of usability experience.

Usability Experience of “Lightweight”						
(I) Wearable Devices	(J) Wearable Devices	Mean Difference (I–J)	Standard Error of Mean	*p* Value	Confidence Interval, CI Lower Limit → Upper Limit	
GPS watch	Bluetooth card	0.775 *	0.236	0.013	0.12	1.44
	Bluetooth bracelet	0.517	0.264	0.298	−0.22	1.26
	Bluetooth pendant	0.648	0.253	0.090	−0.06	1.36
	GPS wrist	0.600	0.249	0.125	−0.10	1.30

* *p* < 0.05.

#### 11.4.2. Analysis of Different “Degrees of Dementia”

In this study, an ANOVA was used to test the differences in each research variable among subjects with different “dementia degrees”. The data were analyzed through the above-mentioned research methods. If the analysis results reached a noticeable difference, then the Scheffe method or Tukey HSD method was used to make post hoc comparisons. The study found that there was no noticeable difference in the results of the pretest (the first test), but there was a prominent difference in the post-test (second test) in the post hoc multiple comparisons. From the results of multiple comparisons after the test, it is worth noting that the older adults with CDR: 2 (moderate dementia) seem to show unsatisfactory results for wearing the device. This result may be caused by some differences in their awareness and understanding of the device. To better render evidence for the device based on the degree of dementia, more time would need to be spent on communicating the reason for doing so, this to older adults. The issue of communication and explanation was not considered in this study.

##### Wearing Results of the First Test

(1)For CDR: 0.5 (the older adults with very mild dementia), wearing the device showed two feelings: “Lightweight” and “Effortless”.(2)For CDR: 1 (the older adults with mild dementia), wearing the device showed five feelings “Simple”, “Relaxed”, “Stable”, “Useful”, and “Comfortable”.(3)For CDR: 3 (the older adults with severe dementia), wearing the device had three feelings: “Convenient”, “Reliable”, and “Safe”.

##### Wearing Results of the Second Test

(1)For CDR: 0.5 (the older adults with very mild dementia), wearing the device showed a “Comfortable” feeling.(2)For CDR: 1 (the older adults with mild dementia), wearing the device had several feelings of “Convenient”, “Reliable”, and “Lightweight” when wearing a device.(3)For CDR: 2 (the older adults with moderate dementia), wearing the device had three feelings of “Relaxed”, “Safe”, and “Useful”.(4)For CDR: 3 (the older adults with severe dementia), wearing the device had two feelings: “Simple” and “Safe”.

##### Results of Multiple Comparisons after Wearing in the Second Test

(1)In terms of the four types of sexual experiences “Simple”, “Relaxed”, “Convenient” and, “Useful”, the sensitivity ratio of the older adults suspected of dementia was better than that of CDR: 2 (moderate dementia older adults).(2)In terms of sexual experience “Stable”, the sensitivity between the older adults suspected of dementia and CDR: 1 (elderly with mild dementia) was better than that of CDR: 2 (elderly with moderate dementia).(3)In terms of usability “Reliable”, the sensitivity between the older adults suspected of dementia and CDR: 1 (mild dementia older adults) was better than that of CDR: 2 (moderate dementia older adults).(4)In terms of usability “Reliable”, the sensitivity of the older adults suspected of dementia, CDR: 1 (elderly with mild dementia) and CDR: 3 (elderly with severe dementia) was higher than that of CDR: 2 (elderly with moderate dementia).(5)In terms of sexual experience “Effortless”, the susceptibility of the older adults suspected of dementia was better than that of CDR: 1 (elderly with mild dementia) and CDR: 2 (elderly with moderate dementia).

### 11.5. Difference Analysis of Different “Field Settings”

In this study, a comparison of the differences was made with these two tests, and the test results of the pre-test (the first test) were evaluated afterwards as “Comfortable”, “Safe”, “Relaxed”, “Convenient”, and “Reliable”, through multiple comparisons. These five points had significant effects.

The reason for this aspect of the result may be the care given by the family when persuading the older adults to wear the device; this may have created an emotional connection, prompting a positive and salient response to the device. The results of the post-test (the second test) were divided into six aspects: “Convenient”, “Stable”, “Reliable”, “Safe”, “Useful”, and “Simple”. Multiple comparisons showed a significant positive response, which may be due to the fact that the older adults then had relevant wearing experience and were less likely to oppose wearing the device. In addition, after the wearing habits established by the pre-test, the older adults had a sense of dependence on the device, which revealed differences in the research results.

#### 11.5.1. Analysis of the First Test

This study adopted an ANOVA test to examine the research variables in different “field settings”. The data analysis was carried out through the research methods mentioned above. If the analysis results reached a significant difference, then the Scheffe method or Tukey’s HSD method would be used to conduct post hoc comparisons. In this study, it was found that different “field settings” did not affect the usability experiences of “Relaxed”, “Convenient”, “Reliable”, “Safe”, and “Comfortable”. There was no remarkable difference in any item, but consumers with different “field settings” showed significant differences in “usability experience”. Using Tukey’s HSD method and the Scheffe method for post hoc comparisons, it was found that in terms of the overall usability experience-related dependent variables, there were several explanations. First of all, in terms of the usability variable “Relaxed”, the usability of “a” was better than that of “g”. Moreover, the usability of “c” was better than that of “a” and “g”. As for the usability variable “Convenient”, the usability of “c” was better than that of “a” and “g”. Meanwhile, regarding the usability experience variable “Reliable”, the usability of “a” was better than that of “g”. Next, concerning the usability variable “Safe”, the usability of “a” was better than that of “a”. In terms of the variable “Safe”, the usability of “c” was better than that of “g”, and the usability of “d” was better than that of “a”. Finally, for the usability variable “Comfortable”, the usability of “c” was better than that of “a” and “g”. The aforementioned day care centers are classified as follows:Douliu “Little Sun” Day Care Center for the older adultsGukeng “Little Sun” Day Care Center for Older Adults“Bulaotang” Dongshi Day Care Service Center“Bulaotang” Taixi Day Care Service Center“Chang Tai Old School” Douli Day Care Center“Chang Tai Old School” Huwei Day Care Center“Chang Tai Old School” Baozhong Day Care Center

#### 11.5.2. Analysis of the Second Test

In this study, an ANOVA was used to test the research variables in different “field settings”. The data analysis was performed through the research methods mentioned above. If the analysis results reached a significant difference, then the Scheffe method or Tukey’s HSD method would be used to conduct post hoc comparisons. The study found that different “field settings” did not affect the usability experience variables “Reliable”, “Convenient”, “Stable”, “Safe”, “Useful”, and “Comfortable”. There was no noteworthy difference in any item, but consumers with different “field settings” showed differences in their “usability experience”. Tukey’s HSD method and the Scheffe method were used to conduct post hoc comparisons, and several explanations were found in terms of the overall usability experience dependent variable. First of all, in terms of the usability experience variable “Convenient”, “e” and “c” received more positive responses than “a”. Next, for the usability experience variables “Stable”, “Reliable”, “Safe”, and “Useful”, except for “a” and “d”, all the other five fields received positive responses. Finally, in terms of the usability experience variable “Simple”, the usability of “c” was the best. 

### 11.6. Test Interview Content

According to the experimental equipment, background operations, and feasibility of this research, interviews were conducted with the social workers and caregivers at the day care centers. This was to understand the feedback from both the older adults and the caregivers. The overall content of the results is organized as follows:

#### 11.6.1. Charging Convenience

Because the standby time of the device was different, and each GPS device had an independent charger, it was more difficult to charge collectively and pay attention to the power.

(1)Feedback 1: “It was exhausting that you have to charge it every day. It would be the best if not to do so. It may be better to change the battery type. It would be hard to promote charging once every 2–3 days because the family members would forget to do so”.(2)Feedback 2: “I would remind the teacher of the time to charge it so that the teacher would pay attention to the battery. It was a bit troublesome at first, but the teacher was getting used to it in the second month.”

#### 11.6.2. The Condition of the Wearable Device

There were various types of devices. Generally, caregivers and social workers tended to recommend Bluetooth cards and Bluetooth charms. Because these two types of device are minimally invasive, it was easier for the older adults to wear them freely. In addition, they are so small that the caregivers and social workers may forget that the older adults are wearing them. However, this may alleviate the chance of the older adults taking the devices off.

#### 11.6.3. Bluetooth Card and Bluetooth Charm

(1)Feedback 3: “Cards and charms are the most acceptable ones, and the rope must be tied short from the back of the neck”.(2)Feedback 4: “The charms and cards are easy to wear. The charms can be placed inside the clothes. It was the most convenient to use and did not interfere with daily life. The cards would readily hang on the chest, causing the elders a bit inconvenience when washing their hands”.(3)Feedback 5: “They would not reject when we put things in their clothes. They rather preferred smaller devices”.

#### 11.6.4. Bluetooth Wristband

(1)Feedback 6: “The cover would separate and fall off, causing the elders to be nervous”.(2)Feedback 7: “The strap of the bracelet was too small, and so I hope the bracelet can be adjustable”.(3)Feedback 8: “Some of them think this device was too tight and uncomfortable to be worn on”.

#### 11.6.5. GPS Watch

(1)Feedback 9: “Male participants preferred GPS watches because this allows them a chance to know what time it is”.(2)Feedback 10: “The GPS watch is too big and heavy, and this reminds me of its existence”.(3)Feedback 11: “Usually, if the elders have the habit of wearing watches, it is acceptable. But if they don’t, the watch should be smaller, because they think it was too big”.

#### 11.6.6. GPS Bracelet

(1)Feedback 12: “The GPS bracelet would push the white device out of the case. The older adults would touch it when washing hands”.(2)Feedback 13: “The use of hook and loop fastener as the adjustment belt makes the elders feel uncomfortable during the wearing process”.(3)Feedback 14: “Some can accept this type, and some would feel that they just do not like putting stuff on their hands”.

#### 11.6.7. Device Background

In most situations, caregivers and social workers recommend the system of LINE, as it is easier to operate and can be applied to both computers and mobile phones, thereby making it easy for caregivers to pay immediate attention to the relevant information.

(1)Feedback 15: “LINE can be viewed on a computer, and it is easier to check on it”.(2)Feedback 16: “I do not have time to look into the APP, but I can see it when LINE shows notifications. I hope there would be (other alert devices) immediate and prominent sound and light prompts, and so the effect would be better”.(3)Feedback 17: “The positioning circle of GPS was pretty wide, which is very inaccurate, except in other counties and cities”.

## 12. Research Conclusions and Research Suggestions

### 12.1. Research Conclusions

Based on the results of this study, the patterns of “wearing devices”, “degree of dementia”, and “field setting” of the older adults showed a significant positive relationship with the results obtained from the post-test. A summary of the different factors associated with the feeling of usability users have in terms of wearing devices was made by referring to the research findings; the results are presented one by one, and the conclusions and suggestions of the research can be used as a practical reference in the future.

### 12.2. Research Suggestions

In this study, the older adults diagnosed with dementia living in Yunlin County, Taiwan, were selected as the research participants. Sample data were collected through a questionnaire, and then the data were subjected to statistical analysis. Based on the relationship model of this research structure, the causal relationship of the overall model can be summarized. The results of the analysis are summarized as follows:

#### 12.2.1. “Dementia Degree” Aspect

In the “dementia degree” aspect, the average number of the low dementia degree group (CDR: 1 or CDR: 0.5) was the highest, and the response was relatively sound, so it can be inferred that the older adults with dementia felt better when wearing the device. It received the highest acceptance rate in this group. This acceptance could be due to the visual appeal of the device attracting the attention of the older adults and stimulating their willingness to wear it.

For the older adults, their experience of the using the device formed a sort of a connection between the device and the family. Although their experience of the device’s usability cannot alter their understanding of the device, it may increase their willingness to wear it and their emotional connection. By correcting the results generated from this survey with particular reference to their acceptance of the device and their experience of the device’s usability, and applying them to future devices, the willingness that older adults have to wear devices can be increased. The reaction of patients with moderate dementia (CDR: 2) was relatively poor. Regarding this aspect, some relatively hidden/invisible devices can be worn by older adults because of the invisibility of the devices.

In the group of patients diagnosed with severe dementia (CDR: 3), these older adults had little concern for things; thus, they did not reject the devices. These patients would be willing to wear these devices.

#### 12.2.2. “Wearable Devices” Aspect

In terms of the aspect of “Wearable devices”, wristband devices were chosen by the majority of patients, and the reviews of these devices were positive.

Through interviews and questionnaires, we learned that because most of the older adults involved in this survey are or were farmers, it was difficult for them to adjust to carrying charm and card-type devices. This is because farmers generally work on the farm, and cannot properly carry devices with them.

Although the card is a common type of device worn by the older adults with dementia, according to interviews, their perceptions of the card were bad. According to the scoring results, wristbands were more prominent, even though it was inconvenient for farmers to wear wristbands. However, in comparison, the convenience and acceptance of wristband devices is relatively high.

#### 12.2.3. “Field Setting” Aspect

In the “field setting” aspect, the average rankings of “Bulaotang” Dongshi Day Care Service Center and Gukeng “Little Sun” Day Care Center were the highest, and their usability experiences were the best, regardless of the different device types.

The acceptance and experience of device usability are related to the activities these older adults carried out in the test field. For example, some of the older adults helped to prepare dishes. During such activities, wearing equipment affects the movement of their hands, which is inconvenient.

Playing games (with educational/rehabilitation-based equipment) installed on the device did not affect the older adults a lot. Therefore, when it comes to the selection of devices to wear, it is necessary to consider the daily activities that the older adults often do, so as to adequately provide the devices they really need.

#### 12.2.4. “Gender” Aspect

In terms of the “gender” aspect, the majority of male subjects were in favor of wearing devices, and their acceptance was relatively high.

According to the results of the interviews, it was found that the older adult women thought it was inconvenient to wear the device. This was because the older adult women tended to participant in more activities, and they were afraid of breaking the device during these activities.

#### 12.2.5. “Interface Management”

Social workers and caregivers preferred LINE more than APP, because LINE can be used on both computers and mobile phones at the same time, the notification texts are retained, and it is easy to check them afterwards.

### 12.3. Research Proposals

This study aimed to discuss the research into the use of alert bracelets by older adults. The conclusions are drawn as follows. These older adults did not have a concept of alert bracelets in their general cognition. The elderly people diagnosed with severe stage dementia had no ability to understand the use of alert bracelets. However, these devices had a significant positive effect on the older adults diagnosed with mild dementia.

Because of the many types of protective bracelets, it is difficult for these family members/caregivers to choose the corresponding protective bracelets these patients need. Moreover, habits and considerations seem to vary from person to person. Therefore, each of the five devices adopted in this experiment may be suggested.

#### 12.3.1. Statistical Calculations

Because some subjects decided to withdraw from the study, the number of subjects decreased, and it was difficult to obtain statistical results.

#### 12.3.2. Device Design

The shape, size, and color of the device need to be further optimized. If a device is used in the form of a wristband, its standby time and battery functions should be extended. The length of the wristband should be more in line with the size of an adult. If the device is used in the form of a sling, its function could be increased by adjusting the length of the sling through which the device could be fixed onto clothing.

#### 12.3.3. Charging Problem

If there is no APP and the charging location of the test device is not easy to find, it is impossible to know whether the device in use is fully charged. In addition, the device needs to be charged once every 2–3 days, and this is difficult for family members to implement. It would be easier to use a replacement battery. Older adults who live alone may consider long-term care centers or community centers to assist them with the task of charging.

#### 12.3.4. Equipment Selection

People with low levels of dementia (CDR: 0.5 or CDR: 1) tend to care about the functions of wearable devices. Accordingly, it was recommended that devices with clear functions, such as watches, be provided, so that the older adults would be more willing to wear them by themselves. Besides, for patients with moderate dementia (CDR: 2) and severe dementia (CDR: 3), it was recommended that hidden/invisible devices such as charms and cards be used, so that it is easier for caregivers to assist them with how to wear them.

#### 12.3.5. Interface Management

In case these older adults go missing, in addition to mobile phone notifications, other assistive reminder functions, such as lighting and sound notifications, can be added, and caregivers can pay greater attention to this information in real time.

In view of the above points (from feedback and suggestions given by caregivers and older adults), if the device can be used without charging and the overall design is light and small, it could be suitable for future promotion and application.

## Figures and Tables

**Figure 1 geriatrics-08-00093-f001:**
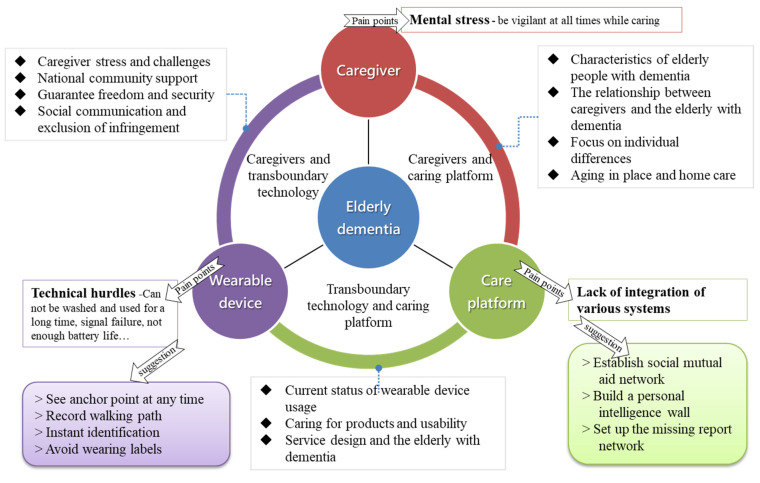
Conceptual framework of wearable devices for elderly dementia patients.

**Figure 2 geriatrics-08-00093-f002:**
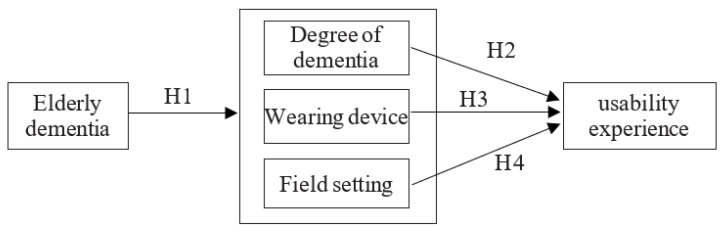
Research experimental framework.

**Table 1 geriatrics-08-00093-t001:** Stages of dementia and typical symptoms and behavior.

Stages	Early	Moderate	Advanced
Symptoms and behavior	Becoming more forgetful of the details of recent events Loss of sense of duration and ability to comprehend how much time has passed Struggling to follow a conversation or find the right word Showing inferior judgement and making poor decisions Showing low task efficiency, losing motivation for work Losing interest in hobbies and activities Having difficulty concentrating; being unable to memorize content just read; being more irritable or upset	Being more forgetful of recent events, addresses, and phone numbers Being confused regarding time and place Talking and calculating become difficult to do Being neglectful of hygiene, cooking or eating circumstances Needing help with dressing and toileting Wandering around streets; seeing or hearing things that are not there	Being incontinent; having trouble with eating, exercising, and moving Showing no recognition of friends and family Losing the ability to understand or use speech Finding it difficult to count to 10 Becoming lost if away from home Showing compulsive behavior, being emotionally unhinged and detached from reality

**Table 2 geriatrics-08-00093-t002:** Statistics on the number of current residents registered.

The Number, Percentage, and Dependency Ratio of the Three-Stage Age Population in Each County and City at the End of the Year											
At the End of the Year (2019) of the Republic of China											Unit: Percentage, %
District Area	Three-Stage Age Population				Year Age Point Match Hundred Point Ratio (%)			Childcare Ratio	Aged Care Ratio	Dependency Ratio	Aging Index
	Total	0–14	15–64	65+	0–14	15–64	65+				
total count	23,603,121	3,010,351	16,985,643	3,607,127	12.75	71.96	15.28	17.72	21.24	38.96	119.82
New Taipei City	4,018,696	486,253	2,953,932	578,511	12.10	73.50	14.40	16.46	19.58	36.05	118.97
Taipei City	2,645,041	355,500	1,811,597	477,944	13.44	68.49	18.07	19.62	26.38	46.01	134.44
Taoyuan City	2,249,037	334,572	1,642,117	272,348	14.88	73.01	12.11	20.37	16.59	36.96	81.40
Taichung City	2,815,261	400,574	2,052,438	362,249	14.23	72.90	12.87	19.52	17.65	37.17	90.43
Tainan City	1,880,906	227,496	1,357,463	295,947	12.10	72.17	15.73	16.76	21.80	38.56	130.09
Kaohsiung City	2,773,198	328,427	2,006,319	438,452	11.84	72.35	15.81	16.37	21.85	38.22	133.50
Taiwan Province	7,067,708	863,131	5,043,626	1,160,951	12.21	71.36	16.43	17.11	23.02	40.13	134.50
Yilan County	454,178	53,627	325,380	75,171	11.81	71.64	16.55	16.48	23.10	39.58	140.17
Hsinchu County	563,933	91,456	401,426	71,051	16.22	71.18	12.60	22.78	17.70	40.48	77.69
Miaoli County	545,459	66,926	388,155	90,378	12.27	71.16	16.57	17.24	23.28	40.53	135.04
Changhua County	1,272,802	163,021	906,565	203,216	12.81	71.23	15.97	17.98	22.42	40.40	124.66
Nantou County	494,112	53,113	352,746	88,253	10.75	71.39	17.86	15.06	25.02	40.08	166.16
Yunlin County	681,306	76,782	478,365	126,159	11.27	70.21	18.52	16.05	26.37	42.42	164.31
Chiayi County	503,113	46,469	357,640	99,004	9.24	71.09	19.68	12.99	27.68	40.68	213.05
Pingtung County	819,184	86,149	591,915	141,120	10.52	72.26	17.23	14.55	23.84	38.40	163.81
Taitung County	216,781	25,127	155,418	36,236	11.59	71.69	16.72	16.17	23.32	39.48	144.21
Hualien County	326,247	38,716	233,265	54,266	11.87	71.50	16.63	16.60	23.26	39.86	140.16
Penghu County	105,207	10,893	76,951	17,363	10.35	73.14	16.50	14.16	22.56	36.72	159.40
Keelung City	368,893	38,171	269,711	61,011	10.35	73.11	16.54	14.15	22.62	36.77	159.84
Hsinchu City	448,803	76,713	315,824	56,266	17.09	70.37	12.54	24.29	17.82	42.11	73.35
Chiayi City	267,690	35,968	190,265	41,457	13.44	71.08	15.49	18.90	21.79	40.69	115.26
Fujian Province	153,274	14,398	118,151	20,725	9.39	77.08	13.52	12.19	17.54	29.73	143.94
Kinmen County	140,185	12,858	108,178	19,149	9.17	77.17	13.66	11.89	17.70	29.59	148.93
Lianjiang County	13,089	1540	9973	1576	11.77	76.19	12.04	15.44	15.80	31.24	102.34
Explanation: Dependency ratio refers to the dependent population (0 to 14 years old and over 65 years old)								Department of Household Affairs, Ministry of the Interior			
A simple measure of the support burden of the working-age population (15–64 years old).											

**Table 3 geriatrics-08-00093-t003:** Wearable devices and descriptions of each item.

Title	Reference Diagram	Active Communication Method Description	Appearance Weight	Wearing Method	Operation Method	Sensing Range/Sensing Distance	Battery Life
Bluetooth card	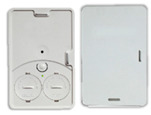	iBeacon Bluetooth positioning device uses D+ card detection, track data upload to the cloud, can query location from computer or mobile phone, can be integrated with health insurance card, one-key SOS emergency help function, lasting use.	Card less than 25 g	Lanyard	Line or web page	Indoors & outdoors, can be sensed distance is 20–50 m, approximate location error is less than 5 m	12 months
Bluetooth bracelet	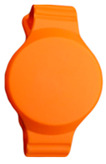	iBeacon Bluetooth positioning device real-time positioning, can check location at any time, record track trajectory, safe electronic fence setting, low battery warning.	Bracelet less than 25 g	Wrist	Line or web page	Indoor & outdoor, can sense distance is 20–50 m, approximate location error is less than 5 m	6 months
Bluetooth pendant	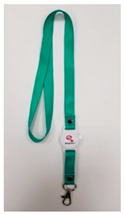	iBeacon Bluetooth positioning device provides received location information, location information history record, out of distance notification family mobile phone.	Label less than 25 g	Lanyard	Line or web page	Indoor & outdoor, can sense distance is 20–50 m, approximate location error is less than 5 m	6 months
GPS wrist	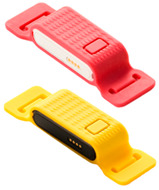	GPS real-time positioning, it is more secure to track the location of the older adults, and you can know the location of the older adults in real time. APP can choose base station mode or GPS mode.	Wristband about 20 g	wrist	Exclusive APP	Outdoors, the positioning accuracy can reach 5–10 m, the approximate position error is about 50–200 m when the reception is good, and 500–1000 m when the reception is poor	Timing positioning standby time up to 4 days
GPS watch	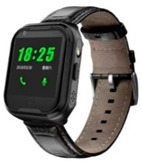	GPS real-time positioning, it is more secure to track the location of the older adults, and you can know the location of the older adults in real time.	Watch about 45 g	wrist	Exclusive APP	Outdoors, the positioning accuracy can reach 5–10 m, the approximate position error is about 50–200 m when the reception is good, and 500–1000 m when the reception is poor	Standby time up to 5 days

**Table 4 geriatrics-08-00093-t004:** The first gender and usability experience *t*-test.

	Gender	*N*	Average Value	Standard Deviation	Standard Error of Mean
Usability experience “Simple”	Men	27	2.26	0.813	0.156
	Women	57	2.33	0.664	0.088
Usability experience “Relaxed”	Men	27	2.56	0.934	0.180
	Women	57	2.21	0.725	0.096
Usability experience “Convenient”	Men	27	2.63	0.688	0.132
	Women	57	2.40	0.776	0.103
Usability experience “Stable”	Men	27	2.37	0.688	0.132
	Women	57	2.26	0.720	0.095
Usability experience “Reliable”	Men	27	2.70	0.542	0.104
	Women	57	2.54	0.600	0.079
Usability experience “Safe”	Men	27	2.56	0.698	0.134
	Women	57	2.46	0.657	0.087
Usability experience “Useful”	Men	27	2.63	0.688	0.132
	Women	57	2.44	0.655	0.087
Usability experience “Lightweight”	Men	27	2.19	0.786	0.151
	Women	57	2.14	0.854	0.113
Usability experience “Effortless”	Men	27	2.37	0.884	0.170
	Women	57	2.35	0.744	0.099
Usability experience “Comfortable”	Men	27	2.33	0.832	0.160
	Women	57	2.46	0.908	0.120

**Table 5 geriatrics-08-00093-t005:** The second gender and usability experience *t*-test.

	Gender	*N*	Average Value	Standard Deviation	Standard Error of Mean
Usability experience “Simple”	Men	25	2.64	0.952	0.190
	Women	50	2.44	0.577	0.082
Usability experience “Relaxed”	Men	25	2.68	0.802	0.160
	Women	50	2.44	0.733	0.104
Usability experience “Convenient”	Men	25	2.76	0.663	0.133
	Women	50	2.58	0.702	0.099
Usability experience “Stable”	Men	25	2.84	0.374	0.075
	Women	50	2.72	0.454	0.064
Usability experience “Reliable”	Men	25	2.84	0.374	0.075
	Women	50	2.76	0.431	0.061
Usability experience “Safe”	Men	25	2.84	0.374	0.075
	Women	50	2.72	0.454	0.064
Usability experience “Useful”	Men	25	2.80	0.500	0.100
	Women	50	2.76	0.591	0.084
Usability experience “Lightweight”	Men	25	2.28	0.542	0.108
	Women	50	2.48	0.789	0.112
Usability experience “Effortless”	Men	25	2.44	0.583	0.117
	Women	50	2.44	0.705	0.100
Usability experience “Comfortable”	Men	25	2.56	0.917	0.183
	Women	50	2.48	0.814	0.115

## Data Availability

Restrictions apply to the availability of these data. Data were obtained from the Yunlin County government and are available from the authors with the permission of the Yunlin County government.

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
