# Peer review of "Service Design of a Loss Prevention Device for Older Adults with Dementia"

_geriatrics, 2023, doi:10.3390/geriatrics8050093_

Round 1

Reviewer 1 Report

Hello, good afternoon, 

Please comment the following data to improve the status of the article or modify if necessary. 

Line 45, very old reference to support the evolution of health cost. 

Line 341, what are the validated instruments to collect the referent information? If they are not validated and have been created ad hoc, their information is not objective and therefore useless for subsequent analysis. 

Line 490, 500, 520, 528, where are the data analysis? 

I understand the objective and interest of the authors for the implementation and development of a digital tool of this caliber but it carries many contraindications or problematic on people with cognitive impairment. These, by the mere fact of deterioration, will not be able to use these devices as they are not digital natives, they have not internalized this behavior and therefore tend to be disinterested and not take advantage of them. 

On the other hand, it involves a lot of training for the main caregiver in charge of the patient's care, it is difficult to know how it is going to be developed. 

From a certain degree of deterioration it will be impossible to measure the usability experience, losing all sense on the part of the user towards the device. 

I see a lot of questions or a lack of data regarding the research if it is assumed that data have been collected with a sample, in principle, scarce to generalize data and provide significant results. 

Author Response

Dear Professor, Hello:
Thanks for your advice,
Relevant revisions for this study are as follows:
-Line 45, very old reference to support the evolution of health cost.
-Correction such as line 41 to 45.

-Line 341, what are the validated instruments to collect the referent information?

-This questionnaire was prepared after discussing with the supervisor. After the pre-test questionnaire, the SPSS software was used to analyze and delete the items. Please see corrections such as lines 341 to 344.

-Line 490, 500, 520, 528, where are the data analysis? 

-The relevant data has been filled in, please refer to Table 4-Table 8.

During the investigation of this study, it is indeed as you said that in the first and second surveys, the cognition of the elderly with dementia has been degraded.
Because of this, this study also interviewed caregivers to let caregivers understand the application of technology, prevent dementia from getting lost, and reduce the pressure on caregivers.

Thank you for your guidance and teaching.

Reviewer 2 Report

The research study investigated the usage of smart alert bracelets among older adults diagnosed with dementia. Through analysis, the study discovered significant positive correlations between factors such as "wearing devices," "degree of dementia," and "field setting" with the outcomes of the post-test. These findings shed light on the acceptance and usability of these devices among older adults with dementia.When considering the degree of dementia, individuals with lower levels exhibited greater acceptance and improved usability when wearing the device. For those with moderate dementia, the suggestion was to utilize hidden or invisible devices to enhance acceptance. On the other hand, patients diagnosed with severe dementia displayed minimal concerns about wearing devices and were more inclined to use them.In terms of wearable devices, wristband devices were the preferred choice among the majority of older adults. This preference was influenced by their agricultural backgrounds, where carrying charm or card-type devices was impractical. Despite the inconvenience for farmers, wristbands were regarded as more convenient and acceptable.   However, there are several significant issues that need to be addressed:  
  1. The absence of case numbers, basic information, and original test data hinders readers from evaluating the author's reasoning and conclusions critically.
  2. Further English revision is required to rectify formatting errors and improve the clarity of the language used in the article.
  3. The article's structure does not adhere to the submission requirements and should be modified accordingly.
  1. Further English revision is required to rectify formatting errors and improve the clarity of the language used in the artcle.

Author Response

Dear Professor, Hello:

Thanks for your advice,

Relevant amendments for this study are as follows:

  1. The absence of case numbers, basic information, and original test data hinders readers from evaluating the author's reasoning and conclusions critically.

      -Relevant data has been filled in, please refer to Table 4-Table 8.

  1. Further English revision is required to rectify formatting errors and improve the clarity of the language used in the article.

     -It has been sent to the English teacher for correction.

  1. The article's structure does not adhere to the submission requirements and should be modified accordingly.

      -Revised in part, please refer to.

Thank you for your guidance and teaching.